# Secure Deep Learning for Intelligent Terahertz Metamaterial Identification

**DOI:** 10.3390/s20195673

**Published:** 2020-10-05

**Authors:** Feifei Liu, Weihao Zhang, Yu Sun, Jianwei Liu, Jungang Miao, Feng He, Xiaojun Wu

**Affiliations:** 1School of Cyber Science and Technology, Beihang University, Beijing 100191, China; ffionliu@buaa.edu.cn (F.L.); weihaozhang@buaa.edu.cn (W.Z.); liujianwei@buaa.edu.cn (J.L.); xiaojunwu@buaa.edu.cn (X.W.); 2School of Electronic and Information Engineering, Beihang University, Beijing 100191, China; jmiaobremen@buaa.edu.cn (J.M.); fenghe@buaa.edu.cn (F.H.)

**Keywords:** metamaterial identification, deep learning, homomorphic encryption, private preserving, terahertz time domain spectroscopy (THz-TDS)

## Abstract

Metamaterials, artificially engineered structures with extraordinary physical properties, offer multifaceted capabilities in interdisciplinary fields. To address the looming threat of stealthy monitoring, the detection and identification of metamaterials is the next research frontier but have not yet been explored. Here, we show that the crypto-oriented convolutional neural network (CNN) makes possible the secure intelligent detection of metamaterials in mixtures. Terahertz signals were encrypted by homomorphic encryption and the ciphertext was submitted to the CNN directly for results, which can only be decrypted by the data owner. The experimentally measured terahertz signals were augmented and further divided into training sets and test sets using 5-fold cross-validation. Experimental results illustrated that the model achieved an accuracy of 100% on the test sets, which highly outperformed humans and the traditional machine learning. The CNN took 9.6 s to inference on 92 encrypted test signals with homomorphic encryption backend. The proposed method with accuracy and security provides private preserving paradigm for artificial intelligence-based material identification.

## 1. Introduction

Metamaterials are artificial materials, designed with special structures, and can exhibit controllable electromagnetic properties [1], which are entirely different from their constituent materials. The broad application of metamaterials in (bio)sensing [2,3,4], imaging [5,6], cloaking [7,8,9,10,11], radar [12], and telecommunications [13] have drawn extensive attention. The development history of metamaterials has experience from equivalent medium metamaterials and surface plasmon metamaterials to information and smart metamaterials [14]. With the continuous efforts of scientists, people have made it possible to digitally encode metamaterials, which means that the big gap between metamaterials and the digital world has been bridged, and intelligent information metamaterials have become an effective bridge between the physical world and the digital world. In the future, metamaterials may become ubiquitous and affect our human being everyday life in many aspects. For example, the imager based on intelligent metasurfaces decorated as a part of wall has already been capable of remotely detecting body movements and restoring high-resolution images of the human body [5]. Other research [10] has designed a multi-wave metasurface carpet cloak, which can hide objects with arbitrary shapes and sizes under electromagnetic, acoustic, and water waves. Meanwhile, intelligent metasurfaces may also have the capabilities in 5G/6G wireless communication systems, where eavesdropping and anti-eavesdropping would become very important. Possessing the ability to change the direction of wave propagation, metamaterials have been used as stealth material especially in military. Although metamaterials have enabled reconfigurability, adaptability, and scalability, and the dawn of their application have already been made great strides, up to now, there are still many interesting physical and practical application researches not yet been explored, such as monitoring and eavesdropping through abusing smart metamaterials, which makes the identification of it an effective way of reconnaissance. As such, this paper proposed a secure and intelligent identification of metamaterials against the potential threat.

For different electromagnetic frequency ranges, metamaterial characterization methods are various. In terahertz (THz) band, time-domain spectroscopy (TDS) is widely employed due to its outstanding performance on extracting both amplitude and phase information. In recent years, rapid advances and considerable application has been witnessed in THz technology such as sensing [15,16], switching [17], modulation [18], and antenna [19]. Besides, numerous scientific publications have been devoted to the use of THz techniques for the detection and identification of materials in recent decades. THz–TDS has been employed as a major probing technique combining with traditional detecting methods to determine the water status incorporated in hydrous minerals [20]. The research of optical parameters of absorption coefficients and refractive index proved that THz–TDS was a promising technique in dehydration analysis. The TDS data were manually analyzed by the positions of absorption peaks or other spectral fingerprints, resulting in time consuming human identification. Until recently, machine-assisted THz technique was reported for the identification of 13 kinds of bi-heterocyclic compounds [21], where features of compounds were extracted from their THz spectra using principal component analysis (PCA), and then classified by the kernel support vector machine (SVM). The system achieved 100% accuracy of the classification of the test compounds, highly surpassing human identification ability.

These previous researches on THz spectra identification evolved from human observation to machine learning methods which mainly required the use of appropriate input features and mathematical apparatus for good performance. There is a tendency to use SVM with PCA in classification, and they indeed achieved perfect accuracy in some cases. However, SVM is based on handcrafted feature engineering. That is, the input features should be carefully selected during preprocessing in order to get good performance. This hand-tuning work is both labor intensive and time consuming, especially for identifying metamaterials whose electromagnetic responses are strongly correlated to many parameters, such as wave incidence angles, polarizations, sample azimuthal angles, geometric structures, and so on.

To this end, deep learning assisted THz identification may offer capabilities, since it is currently the widespread state-of-the-art pattern recognition method. With the historical opportunities brought by big data and hardware acceleration, unprecedented breakthroughs have been made in the field of computer vision. In the task of classification on ImageNet [22], the convolutional neural networks (CNN), i.e., the AlexNet [23], VGG [24], GoogLeNet [25] and ResNet [26] ushered in a remarkable breakthrough, and to some extent, have settled the feature extraction problem in computer vision.

Despite the recent progress on accuracy, private concerns raise because of the sensing data exposed to artificial intelligence (AI), which hinders the application of AI-based identification [27]. If we provided an online metamaterial identification service and other labs attempted to use it, they would need to upload sensing data to our server, resulting in the leakage of both sensing data and identification results. One possible solution is differential privacy [28], where one can determine the amount of data leaked by a single record. However, the concept is useless in the application stage since we are interested in individual record. Another option is the federated learning which uses a more secure aggregation protocol, secure multi-party computing, and the federated average algorithm to train a model without revealing data [29,30]. Nevertheless, the server still has to use sensing data in plaintext during the application stage. One promising solution to these problems is homomorphic encryption (HE) [31], an algorithm to perform computation on encrypted data in the application stage, which is highly practical as basic extensions of privacy-preserving [32]. HE supports essential addition and multiplication operations in CNN. Using HE, the data owner can employ the public key to encrypt his data, send them to the identification server who has no access to the secret key, and finally receive the results in ciphertext.

Inspired by the potential threat of the abuse of metamaterials, in this work, we investigated the secure application of AI techniques to THz identification, exemplified with the identification of metamaterial in mixtures. Through recording THz electromagnetic responses from metamaterial mixed samples, a large number of signals with or without metamaterial were extracted and augmented. A crypto-oriented CNN with HE-backend was applied and we achieved a remarkable accuracy of 100%. 

The innovation of this article lies in (1) this work found the gap in the field of metamaterial identification and filled it with the advanced deep learning method. (2) Considering the application of proposed identification method, a crypto-oriented CNN with HE backend was developed to provide secured identification service. The primary goal of this work is to provide practical AI method for THz signal identification. To the best of our knowledge, this is the first study on the secure use of AI for THz signal identification.

## 2. Materials and Methods

The workflow of private preserving THz metamaterial identification is shown in Figure 1. First in THz-TDS system, the THz wave passed through two lenses, focused onto samples to get the electromagnetic response signals. To meet the need of big data, random augmentation was adopted according to the possible noises. Then, fast Fourier transformation was employed to convert these augmented signals to frequency-domain as the input of CNN. In the training stage, CNN can learn discriminative features through minimizing loss and updating parameters. Once the network was trained well enough to identify the existence of metamaterial, parameters in the model would be exported for application. Afterwards, the model was ready for a private call in application stage, where one encrypted the original data, fed into the network and got the results back in ciphertext, which can only be decrypted by oneself.

### 2.1. THz Measurement

In the data acquisition process, a home-built THz-TDS driven by a femtosecond fiber laser was employed [2]. The THz wave radiated from a commercial photoconductive antenna. In the first place, a femtosecond fiber laser pumps an InGaAs photoconductive antenna to generate a horizontally polarized THz pulse, which was focused onto the sample through two lenses. The transmitted THz pulse after the sample passes through both lenses to the receiving antenna. In this experiment, we collected the spectra of 66 samples, among which 32 contained metamaterials and 34 without metamaterials. For samples with metamaterial, we changed the sample azimuthal angle randomly from 0° to 180°, and added some background materials such as glucose, lactose, and medicines (Vitamin B, ibuprofen, and cimetidine). The metamaterial design parameters are the same in this literature [2]. Fourier transformation was employed to turn the probed temporal waveform signals into frequency-domain, and the amplitudes were input into CNNs to get the binary classification results.

The existence of metamaterials in the mixture is very challenging for humans to distinguish because many factors, such as wave incidence angles, polarizations, sample azimuthal angles, and geometric structures may lead to overlapping absorption peaks in the spectrum. Here we take the mixture of metamaterial and lactose as an example. As is illustrated in Figure 2a, for a specified compound, lactose, all curves with and without metamaterial have similar trends, featuring large interclass similarity. Furthermore, all mixture curves with different azimuthal angles present subtle varieties, indicating great intra-class variances. For example, both blue and orange curves represent the existence of metamaterial with different azimuthal angles, while the orange curve has higher amplitude between 0.75–1.2 THz than that from the blue one. Based on the aforementioned discussion, it is known that the similar resonant features for both the intentionally designed THz metamaterials and the mixed sample materials make the manual analysis by human alone very difficult. A more advanced method is highly demanded.

### 2.2. Data Augmentation

Deep learning benefits from big data, but it is prohibitively expensive to experimentally collect large-scale databases in terms of time and labor cost. Alternatively, data augmentation is convenient to extend the dataset without extra cost. The straightforward approaches for image augmentation include random flipping and random cropping, which are widely used in image identification and validated to be effective. While the aforementioned augmentations were designed for two-dimensional RGB images, which were not applicable to one-dimensional single-channel THz data.

We considered two possible ways to expand the data. First, random perturbation was added to the original time-domain data to simulate system noises. Considering that the data ranged from −9^−9^ to 10^−8^ A, three kinds of Gaussian noises with mean value 0 and variance 10^−10^, 10^−9^, 5 × 10^−9^ were added. The second method to enlarge data scale was simulating the power reduction of the THz source by attenuating the raw time-domain data to 70%, 80%, and 90%. Figure 2b shows the augmentation results, where one spectrum was augmented to seven with different noises and attenuations. With these augmented data, the diversity of training signals was enhanced, and, therefore, the generalization ability of AI model can be improved. We divided the augmented dataset into training set and test set using 5-fold cross-validation.

### 2.3. Crypto-Oriented CNN Design

The augmented Fourier transformed data was the input of the CNNs, the ability of which to construct abstract features makes it well suited to metamaterial identification. To design a crypto-oriented CNN, the depth of CNN should be constrained to prevent accumulated noises in HE decryption, and the ReLU [33] activation should also be replaced due to the limited nonlinear operation supported by HE. Our model comprised two convolutional layers, followed by a fully connected layer. After each convolutional layer, square activation was chosen to improve the ability of nonlinear expression, also alleviate the problem of gradient disappearance. Square activation is defined as $\mathrm{Square}(x)=x^{2}$. Details of CNN are exhibited in Figure 3. Convolutional layer processed the input signal, a one-dimensional array with 61 numbers, by convolving it with a bank of kernels. The shape of kernel to each convolutional layer was (32,3) and (8,3), respectively. Both convolution operations were represented in Figure 3 as cones and had a stride of 2. Subsequently, the red square activation operation worked to provide nonlinear modeling for network. Then, the output of last square activation was flattened to transform the feature to a one-dimensional shape. Finally, the fully-connected layer mapped distributed feature representation to output space. The pooling function was not adopted since max-pooling is not supported by HE and average-pooling was proved inadequate in experiments.

To be precise, assuming *l* to be the number of layers. Before training, weight and bias of each neuron should be initialized as random values *W* and *b*. The data-label pairs were fed through the network as training samples to generate prediction. For input tensor *z^l^*, the convolution process ’*’ resulted in tensor *z^l+1^*, as the input for next convolutional layer.
(1)$$z^{l+1}=\mathrm{Square}(z^{l}*W^{l+1}+b^{l+1})$$

In the last fully-connected layer, the unnormalized probabilities (aka logits) *o* was obtained and regarded as the confidence of metamaterial existence.
(2)$$o=z^{l}W^{l+1}+b^{l+1}$$

Then, sigmoid function was performed to obtain predicted class labels $\hat{y}$. The errors between $\hat{y}$ and ground truth labels *y* were calculated using binary cross-entropy (BCE) loss function [34,35], and then back-propagated through the network employing the chain rule. The above iteration would stop when training epoch is up to 100. Sigmoid scaled each component in the interval (0,1), thus can be interpreted as probabilities. As a nonlinear function, sigmoid would not participate in the HE operation in application stage. The definition of sigmoid is shown as below.
(3)$$\hat{y}=\mathrm{sigmoid}(o)=\frac{1}{1+\exp(-o)}$$

The goal of training CNNs was to minimize the error between $\hat{y}$ and *y*. To penalize non-matching cases, BCE error function of batch *m* is defined to better align network outputs and targets.
(4)$$L(\hat{y},y)=-\frac{1}{m}\sum_{i=1}^{m}\left(y_{i}\log{\hat{y}}_{i}+(1-y_{i})\log(1-{\hat{y}}_{i})\right)$$

After that was the back-propagation, which aimed to update parameters by computing partial derivative from the output layer to the input layer.
(5)$$\frac{\partial L({\hat{y}}_{i},y_{i})}{\partial W}=\sum_{i=1}^{m}\mathrm{prod}(\frac{\partial L({\hat{y}}_{i},y_{i})}{\partial o},\frac{\partial o}{\partial W^{i}})$$

Function “prod” returns the product of all the values present in its arguments. The performance of the network was improved by approximately minimizing the training objective. Root mean square propagation (RMSprop) [36], an adaptive learning rate method proposed by Geoff Hinton, was expected to address sharp decline in learning rate. For parameter *θ_t_* at time *t*, the new *θ_t+1_* was obtained.
(6)$$\theta_{t+1}=\theta_{t}-\frac{\eta }{\sqrt{E{\left[g^{2}\right]}_{t}+\varepsilon }}g_{t},$$
(7)$$E{\left[g^{2}\right]}_{t}=\gamma E{\left[g^{2}\right]}_{t-1}+(1-\gamma )g_{t}^{2},$$
where the learning rate *η* was suggested to be 0.001 and *g_t_* denoted the gradient at time *t*. The denominator was the decaying average of the root mean square of the gradient. Adding momentum *γ* made the speed on the dimension with constant gradient go faster and the changed gradient go slower, so it could accelerate convergence and reduce oscillation.

### 2.4. Private Preserving Application

Figure 4 shows how the HE enables a client to implement AI identification on confidential THz signals using a remote, untrusted server. Client encrypted his data using public key *pk* and sent it to remote server, which was received and fed into crypto-oriented CNN model. Afterwards, the client decrypted the output of CNN using secret key *sk* and performed sigmoid function to obtain the final results. In no case should the server gain access to the existence of metamaterial.

Our fully homomorphic encryption scheme is based on the assumed hardness of the Ring Learning with Errors problem [37], whose parameters contain *N* as the polynomial modulus degree. It is necessary to select parameters of sufficient size so that the amplification of random noise will not make the original message unrecoverable. Therefore, the multiplication times on the ciphertext should be no more than *L*, the maximum multiplicative depth. In experiment, we chose *N* = 2^13^ and *L* = 8. Privacy was guaranteed through four algorithms:**KeyGen**, a randomized algorithm that takes a security parameter *λ* as input, generates some representations of a finite ring *R* with addition operator ⊕ and multiplication operator ⊗, and outputs a *sk* and *pk*.**Enc**, a randomized algorithm that takes *pk* and a plaintext *π* as input and outputs a ciphertext *ψ* ∈ *R*.**Dec** takes *sk, ψ* as input and outputs the plaintext *π*.**Eval** is an efficient algorithm which takes *pk*, ring *R* and a tuple of ciphertexts $\psi =\{\psi_{1},\ldots ,\psi_{t}\}$ as input, and outputs a ciphertext ψ ∈ R.

For any plaintext *π*_1_, *π*_2_ ∈ *R*, we have
(8)$$\mathrm{Dec}(\mathrm{Enc}(\pi_{1}))⊕\mathrm{Dec}(\mathrm{Enc}(\pi_{2}))=\pi_{1}+\pi_{2},$$
(9)$$\mathrm{Dec}(\mathrm{Enc}(\pi_{1}))⊗\mathrm{Dec}(\mathrm{Enc}(\pi_{2}))=\pi_{1}\times \pi_{2},$$
where + and × are the standard addition and multiplication operations in the ring *R*. The correctness of scheme is defined as
(10)$$\mathrm{if}\;\psi \leftarrow \mathrm{Eval}(pk,R,\psi ),\mathrm{then}\;\mathrm{Dec}(sk,\psi )\to R(\pi_{1},\ldots ,\pi_{t}).$$

## 3. Results and Discussion

The CNN model was implemented by the deep learning framework Keras with TensorFlow backend on a workstation equipped with one Intel Core i7-6700 3.40 GHz Processor (64 GB memory) and one NVIDIA TITAN RTX GPU (24 GB graphic memory).

Figure 5 is the test process taking one-fold as an example, where accuracy quickly arrived at 100%, indicating the model’s robustness and ability in predicting. Compared with the red line, data augmentation also has a remarkable effect in enhancing stability.

We conducted the comparison experiment following the mainstream SVM algorithm. The method consisted of two stages. First, PCA was utilized to extract relevant features from a set of observed spectra, and then as input of the SVM with Gaussian kernel to classify the features. Considering the 61 neurons in one input tensor, we chose principal components from 5 to 60.

To prove the difficulty of distinguishing metamaterial according to its THz spectrum, we carried out an experiment on 50 people as human baseline. These people, randomly divided into five groups for each fold test, were asked to figure out which spectrum belongs to metamaterial. Figure 6a illustrates the result as follows. Inevitably, due to the anisotropy of metamaterial in mixture, the spectra are too hard to differentiate by human eyes, achieving the mean accuracy of only 56.95%. Traditional SVM method performed better, with mean accuracy of 87.9%. However, deep learning method CNN has the most outstanding performance, with the accuracy of 100% on every fold.

Frequency-domain spectrum of pure α-lactose monohydrate and its corresponding ciphertext are shown in Figure 6b. HE worked by introducing random noises to ensure privacy. Noises ranging from −4874.36 to 4013.41 are larger than the signals by exorbitant orders of magnitude, which is efficient enough to prevent information leakage.

When security is applied to sensitive data, a balance must be found between accuracy and computational complexity. Different approaches present different trade-offs in terms of accuracy and speed. Though outperforms as the fastest algorithm, SVM does not support any encryption and has poor accuracy. Several HE approaches with different parameters, *N* and *L*, all achieved the accuracy of 100%, among which we figured out the one with the shortest time and least computation, featuring *N* = 2^13^, *L* = 8 and time of 9.6 s on a batch of 92 signals. Even though runtimes increased after employing HE, we must note that runtimes are independent of batch size since computation graphs provide a mechanism for parallel identification. Batching only increases throughput significantly.

## 4. Conclusions

In summary, we identified the existence of metamaterial in mixtures using THz technique and crypto-oriented CNN model. The feasibility of CNN has been demonstrated against the traditional and extensively used machine learning-based approaches, since it can be trained by the raw signals to learn discriminative features. The main contribution of this paper is to implement identification of metamaterials in mixtures which is a challenging task for human and SVM. With the assistant of AI, metamaterials can be successfully identified with high efficiency. The superiority was proved by evaluating the performance with human beings and SVM, where the classification accuracy on test set is obviously improved. Furthermore, HE was integrated for security purpose, thus, the private preserving identification service can be applied in client-server model. Although we assumed a certain structure, our method is applicable to other configurations of metamaterials only with the network structure fine-tuned.

Our work demonstrates the applicability of AI to the THz recognition field. Under the premise of sufficient data, AI will open up a new research path for the THz identification of different materials. With the improvement and popularization of THz technology in the future, AI will be closely combined with THz technology in the fields of (bio)sensing, imaging, cloaking, and 5G/6G wireless communication. Meanwhile, numerous sensitive THz detectors that can collect the electromagnetic data of celestial objects in the THz band have emerged. This advanced work may shed light in various AI-based applications, such as THz astronomy, security, and smart sensing. We also hope that AI and THz technology will cross over into better results in the future.

## Figures and Tables

**Figure 1 sensors-20-05673-f001:**
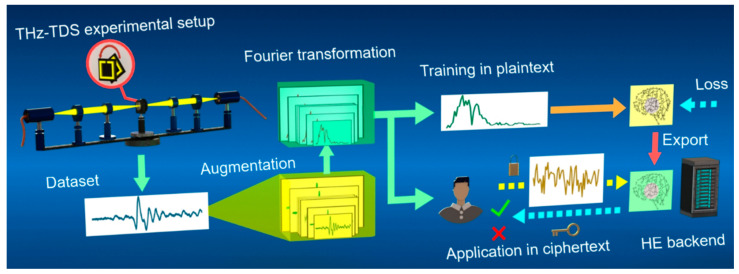
The workflow of private preserving terahertz (THz) metamaterial identification. Photocurrent signals of samples were obtained from THz measurement system and then transformed to frequency-domain. The training stage in plaintext and the application stage in ciphertext are illustrated in the upper right and lower right parts, respectively.

**Figure 2 sensors-20-05673-f002:**
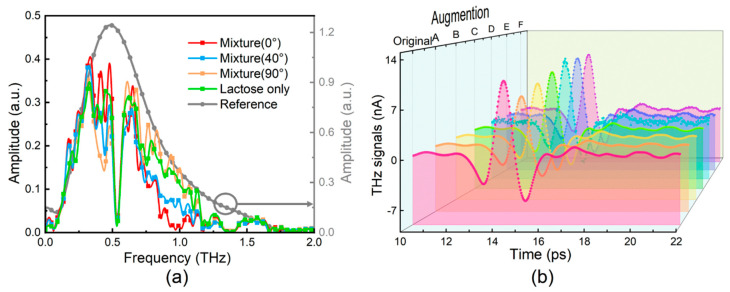
(**a**) Frequency-domain spectra of metamaterial-lactose mixtures with different azimuthal angles of 0°, 40°, 90°, respectively. The grey dotted line is the reference signal without any samples; (**b**) Data augmentation on THz temporal waveform signals. A-F represent signals with attenuation of 70%, 80%, 90%, and signals adding Gaussian noises with mean value 0 and variance 5 × 10^−9^, 10^−9^, 10^−10^, respectively.

**Figure 3 sensors-20-05673-f003:**
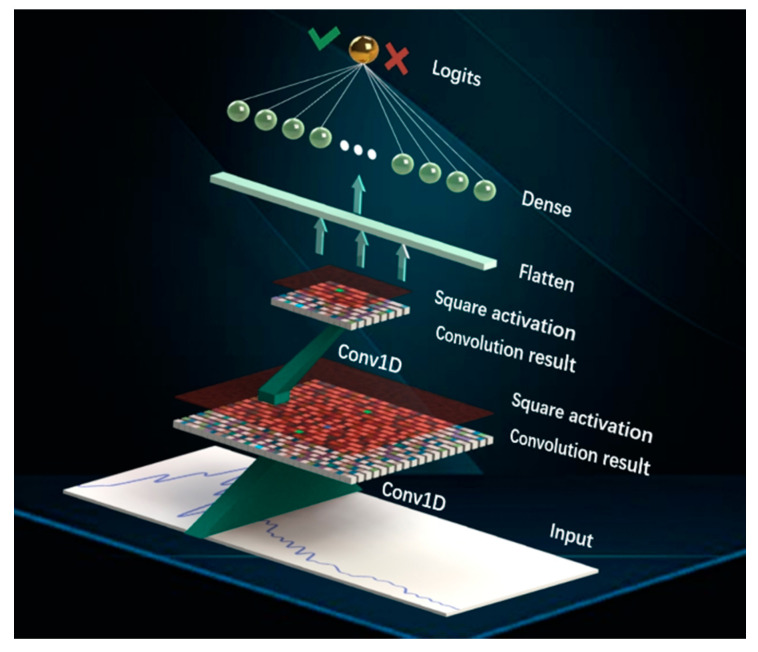
Convolutional neural network (CNN) inferences on raw frequency-domain spectrum using two convolutional layers and one fully-connected layer. “Input” layer was a vector of 61 numbers. In the first “Conv1D” operation, which was represented as a green cone, input signals were convolved by 32 kernels with shape 1 × 3. Then the 30 × 32 shaped output was obtained and represented as a black-and-white grid. Subsequently, the “Square activation” operation was added as a light red square.

**Figure 4 sensors-20-05673-f004:**
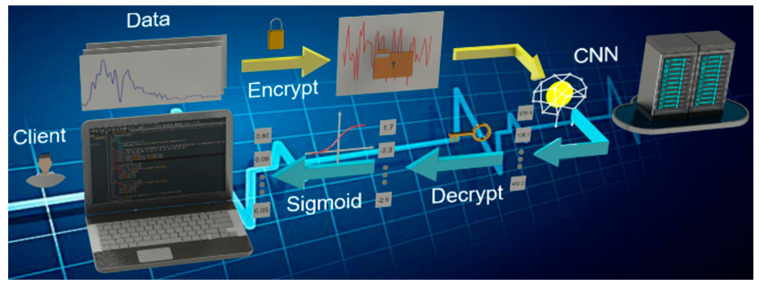
Secure application in client-server model. Possessing private data waiting for identification, client encrypted his plaintext using public key for ciphertext, which was then received by the server. Without decrypting it, the server performed calculations of CNN with homomorphic encryption (HE) backend directly and sent the decrypted results to the client, who uncovered results with his secure key.

**Figure 5 sensors-20-05673-f005:**
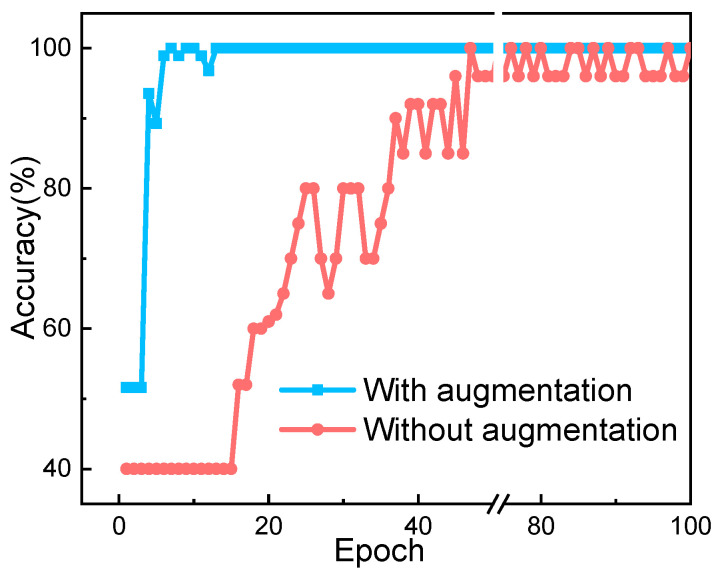
Comparison of accuracy on test set with and without augmentation in one-fold, where augmentation rendered the convergence faster and smoother.

**Figure 6 sensors-20-05673-f006:**
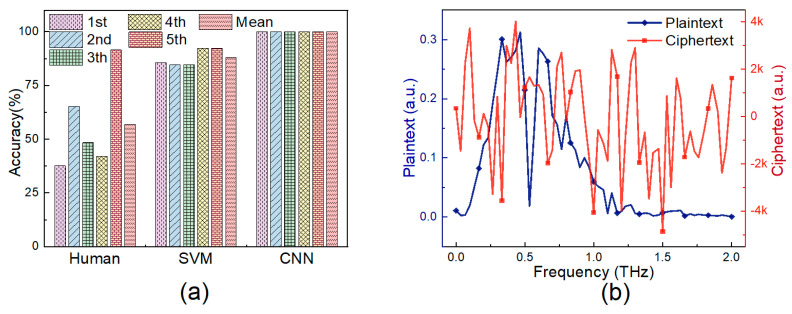
(**a**) Comparison of accuracy on each fold for human, support vector machine (SVM) and CNN; (**b**) Plaintext and ciphertext of one THz signal.

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
