# Peer review of "Secure Deep Learning for Intelligent Terahertz Metamaterial Identification"

_sensors, 2020, doi:10.3390/s20195673_

Round 1

Reviewer 1 Report

Dr. Feifei Liu and the coauthors reported a methodology for the material identification based on a deep learning technology with a secure signification. The authors succeeded to identify the existence of a metamaterial from samples of mixture of alpha-lactose with a higher score than humans and with a secure manner. Such identification technology is very important for applications of the THz wave. However, I have some questions and comments. If these are cleared, this manuscript will be ready for publication.

  1. I’m wondering why the authors assumed the situation to identify a metamaterial from the mixture. I could understand the importance of the material identification technique. But why is the identification of metamaterials necessary?
  2. In line 129, the authors mentioned “it is extremely difficult to identify metamaterial in the THz band by manual analysis alone, and a more advanced method is needed to solve this problem.” Usually, for material identification, one may make a calibration curve for the material at a certain frequency. In metamaterial case, why such approach is difficult?
  3. The authors assumed a certain structure for the metamaterial. Can their method be applied for the identification of arbitrary configurations of metamaterial, in which the spectral shapes depend on the configuration?
  4. In line 146, the authors mentioned “The second method to enlarge data scale was simulating oblique incidence by attenuating the raw time-domain data to 70%, 80% and 90%.” In general, not only the amplitude, but also phase and period can be changed by the incidence angle. Namely, the spectral shape of metamaterials strongly depends on incident angle. Why did the author assume only the amplitude change?
  5. In Fig.2(a), the authors should also show the spectrum of the back ground (without metamaterials and lactose).
  6. In Fig.2(b), the authors showed six augmentation data. It is better to disclose what kinds of fluctuation were introduced in each augment operation with the actual values of parameters. Is there any restriction for the range of the parameter fluctuation?
  7. 3, I could not catch what is depicted at the part between “input” layer and “square activation Conv 1D” layer.
  8. In Eq.(6), an operator of “prod(*,*)” is not defined.

Reviewer 2 Report

The authors have investigated the secure application of AI techniques to THz identification. I found the manuscript very interesting. The manuscript is well written and the results are clearly presented. The authors have fully verified the concept in the experiment. Overall, I am highly supportive of the publication of the work provided that the authors address the following comments.

1-Some of the claims in the manuscript should be toned down a little bit. An example of these claims is the following statement: “This pioneer work may shed light in various AI-based applications such as THz astronomy, security, and smart sensing”. The authors should avoid using overstatement terms such as “pioneer”. In addition, they should discuss more rigorously how their structure can have an application in astronomy?

2- The authors verify the performance of their system in the experiment. However, there is no data in the manuscript that validates the experimental measurements itself! Would that be possible to cross-validate the experimental results with simulation?

3- The introduction does not provide even one sentence about the recent advance of THz technology. There are a lot of interesting applications for THz waves demonstrated in recent years, including sensing: IEEE Sensors Journal11 (2016): 4338-4344, Optics Communications 371 (2016): 9-14, modulation: Journal of Infrared, Millimeter, and Terahertz Waves 34.1 (2013): 1-27, 1-10 switching: IEEE Transactions on Terahertz Science and Technology 5.5 (2015): 725-731 and antenna: Journal of Nanophotonics 10.3 (2016): 036005.

4-The authors did not provide any comment regarding the efficiency of their system. It is true that in simulation, one can easily excite the structure via a port, but the story in practice is rather different. Actually, to the best of my knowledge, most of the terahertz sources suffer from very low values of conversion efficiency (below 10%).This may reduce the practical worth of their configuration considering the current technology. I would suggest the authors to at least write some sentences regarding this limitation in their article.

5- The relation given in Eq. 1 does not deserve to be a separate equation with a specific number. The authors could include it in the main text.

6- Captions of the figures are not good. The authors should provide an overview of the whole figure in the caption. For instance, in Fig.4, they would better describe what the system is and what it is composed of, etc.  

7- While the paper is well written there are still several language glitches in the paper such as various AI-based application (instead of applications). Please proofread the entire paper.

Reviewer 3 Report

The paper presents interesting idea but it require some improvements and clarifications.

First, the main contribution of this paper should be outlined. The authors stated that it is the first approach of using secure AI to Thz identification. But it does not mean that the paper presents a new results. Thus, it should be outlined what new results where obtained and what are the advantages of the presented method in comparison with existing ones.

Next, the motivation to this paper is not clear. It should be explained if the main goal is to improve the efficiency of the THz signal identification or to support  security. Both concerns are quite different and it is important what is the primary goal of this approach.

Since the work is the first approach to apply CNN to THz identification some discussion concerning future work would be valuable. 

Round 2

Reviewer 2 Report

The authors have addressed my previous comments.